# Gut-Kidney Impairment Process of Adenine Combined with *Folium sennae*-Induced Diarrhea: Association with Interactions between *Lactobacillus intestinalis*, *Bacteroides acidifaciens* and Acetic Acid, Inflammation, and Kidney Function

**DOI:** 10.3390/cells11203261

**Published:** 2022-10-17

**Authors:** Xiaoya Li, Xinxin Peng, Bo Qiao, Maijiao Peng, Na Deng, Rong Yu, Zhoujin Tan

**Affiliations:** 1School of Traditional Chinese Medicine, Hunan University of Chinese Medicine, Changsha 410208, China; 2Department of Pediatrics, The First Affiliated Hospital of Hunan University of Chinese Medicine, Changsha 410208, China; 3School of Pharmacy, Hunan University of Chinese Medicine, Changsha 410208, China

**Keywords:** gut microbiota, adenine combined with *Folium sennae*-induced diarrhea, characteristic bacteria, SCFAs, intestinal inflammatory response, kidney function

## Abstract

Background: Extensive evidence suggests that gut microbiota may interact with the kidneys and play central roles in the pathogenesis of disease. However, the association of gut microbiota-kidneys in diarrhea remains unclear. Methods: A diarrhea mouse model was constructed by combining adenine with *Folium sennae*. We analyzed the characteristics of the gut content microbiota and short chain fatty acids (SCFAs); and explored the potential link between gut content microbiota, SCFAs, intestinal inflammatory response and kidney function. Results: Characteristic bacteria *Lactobacillus intestinalis* and *Bacteroides acidifaciens* were enriched in the gut contents of mice. The productions of SCFAs were remarkably inhibited. Model mice presented an increased trend of creatinine (Cr), interleukin-6 (IL-6) and tumor necrosis factor-α (TNF-α), a decreased trend of blood urea nitrogen (BUN) and secretory immunoglobulin A (SIgA). The pathological analysis proved obvious damage to the kidney structure. *Lactobacillus intestinalis* and *Bacteroides acidifaciens* exisited in the correlations with acetic acid, intestinal inflammatory response and kidney function. Conclusions: Adenine combined with *Folium sennae*-induced diarrhea, altered the structure and function of the gut content microbiota in mice, causing the enrichment of the characteristic bacteria *Lactobacillus intestinalis* and *Bacteroides acidifaciens*. The interactions between *Lactobacillus intestinalis*, *Bacteroides acidifaciens* and acetic acid, intestinal inflammation, and kidney function might be involved in the process of gut-kidney impairment in adenine, combined with *Folium sennae*-induced diarrhea.

## 1. Introduction

Diarrhea, a common and frequent disease worldwide, accompanies the development of diabetes, tumors, ulcerative colitis and other diseases, and seriously affects the physical and mental health of patients and their quality of life [1,2,3]. The Kidney-Yang Deficiency Syndrome, as one of the common clinical syndromes of diarrhea in traditional Chinese medicine (TCM), is still challenging to study, due to the presence of complex pathogenic factors [4]. The gut microbiota has long been considered an important player in maintaining human health, and disorders of microbiota cause the development of many diseases [5]. Studies have proven that the occurrence of diarrhea is closely related to gut microbiota disorders [6]. Diarrhea is often followed by the disruption of the gut microbiota, and there were significant differences in the composition of the gut microbiota between patients with diarrhea and those without [7]. Furthermore, short chain fatty acids (SCFAs), metabolites of the gut microbiota, have a positive impact on the maintenance of intestinal health. Among them, acetic acid, propionic acid, butyric acid, valeric acid, isobutyric acid and isovaleric acid are essential for regulating the homeostasis of the intestinal environment, protecting the barrier function of the intestinal mucosa, balancing the energy metabolism of the intestinal epithelial cells and countering the inflammatory response [8,9,10,11,12]. Xu et al. indicated that acetic acid, propionic acid, butyric acid, isobutyric acid, valeric acid and isovaleric acid found in the feces of children with diarrhea, were significantly lower than those in the feces of healthy children, suggesting that an abnormal production of acetic acid, propionic acid, butyric acid, isobutyric acid, valeric acid and isovaleric acid were key factors in the development of diarrhea [13]. Therefore, studying the changes in the gut microbiota and SCFAs during diarrhea, helps us to understand the underlying pathophysiological mechanisms.

With the upgrading of sequencing technology and bioinformatics analysis, the regulatory functions of gut microbiota have been heavily explored [14,15]. Gut microbiota and SCFAs not only directly affect gut health, but also play a targeted regulatory role on the distal organs, such as the brain, liver and kidney [16]. Taking chronic renal failure (CRF) as an example [17], *Lactobacillus*, *Bifidobacterium*, *Enterococcus faecalis* and *Enterococcus faecalis* showed significant changes in the gut microbiota of patients with advanced CRF. The correlation analysis confirmed that BUN and Scr levels in patients, were negatively correlated with *Lactobacillus* and *Bifidobacterium* (*p* < 0.05) and positively correlated with *Enterococcus faecalis* and *Enterococcus faecalis* (*p* < 0.05). It was suggested that the changes of the gut microbiota in advanced CRF patients were closely related to kidney function. Additionally, Liu et al. found an abnormal production of acetic acid, propionic acid, butyric acid, isobutyric acid, valeric acid and isovaleric acid in dogs with chronic renal failure that were positively correlated with kidney dysfunction [18]. It was evident that gut microbiota disorders and an abnormal production of SCFAs were closely associated with kidney impairment. Thus, we hypothesized that the pathogenic mechanism caused by the gut-kidney impairment in adenine, combined with *Folium sennae*-induced diarrhea, might be related to the disturbance of the gut microbiota and the abnormal production of SCFAs, but the exact mechanism of action still needs further study.

Studies have pointed out that certain doses of adenine could impair kidney function to a certain extent and also affect the body’s energy metabolism [19]. *Folium sennae* is a bitter-cold laxative, commonly used in TCM. In the preliminary experiments, we investigated the effects of different doses of adenine modeling on the kidney function and gut microbiota of mice. We analyzed the correlation between the gut microbiota and the kidney function during modeling and elucidated the role of gut microbiota in the process of kidney dysfunction after adenine modeling. Among them, adenine (50 mg/(kg·d) for 14 days by gavage) damaged the kidney structure and the function of mice, and caused disorders of the gut microbiota, and characteristic bacterium *Lactobacillus hamsteri* affected the process of the kidney functional impairment. Therefore, the adenine-induced kidney impairment is not an independent event. There is an association between kidney impairment caused by the adenine modeling and the disturbance of the gut microbiota. Our research group’s preliminary research presented that mice showed obvious diarrhea symptoms and caused gut microbiota disorders after the *Folium sennae* modeling [20]. Furthermore, we compared the effects of adenine combined with *Folium sennae* at different doses and days, on the kidney and gut functions in mice and presented that adenine (50 mg/(kg·d), gavaged for 14 days), combined with *Folium sennae* (10 g/(kg·d), gavaged for 7 days,) significantly caused impairment of the kidney and gut function in mice [21]. Subsequently, we have successfully constructed and validated a mouse model of diarrhea using the same modeling method as described above, thus confirming the reliability of the model [22]. Based on these, we used adenine combined with *Folium sennae* to construct a diarrhea mouse model with Kidney-Yang Deficiency Syndrome, and detected the indicators of the kidney function and intestinal inflammatory response, the gut content microbiota, acetic acid, propionic acid, butyric acid, isobutyric acid, valeric acid and isovaleric acid in mice. The purpose of the study was to analyze the characteristics of the gut content microbiota in diarrhea, to explore the correlation between the characteristic bacteria, SCFAs, the inflammatory response and the kidney function, and to study the role of the gut content microbiota in gut-kidney impairment with adenine combined with *Folium sennae*-induced diarrhea. Our findings provided a new perspective on the range of substances that cause the gut-kidney association in the process of diarrhea, laying the foundation for subsequent causal associations in the studies of specific mechanisms.

## 2. Materials and Methods

### 2.1. Medicine

Adenine (Changsha Yaer Biology Co., LTD, Changsha, China, number: EZ2811A135). *Folium sennae* (Anhui Puren Traditional Chinese Medicine Yinpian Co. LTD, Haozhou, Anhui, number: 2005302). Adenine suspension preparation: adenine was prepared in sterile water to a concentration of 5 mg/mL in proportion to the concentration of the suspension and was prepared daily, as needed [23]. *Folium sennae* decoction preparation: we placed *Folium sennae* in a container with the appropriate amount of water for 30 min. Then, we poured off the water, added 5 times the amount of herbs to the container and boiled for 30 min. We filtered out the liquid by laying sterile gauze flat in a funnel. The filtered dregs were then added to an appropriate amount of water and the decoction was continued by boiling for 15 min. The two decoctions were mixed and then boiled for 15 min. The decoction was concentrated to a concentration of 1 g/mL of raw herbs and stored in a refrigerator at 4 °C [24].

### 2.2. Reagents

Interleukin-6 (IL-6) enzyme-linked immunosorbent assay (ELISA) Kit (Jiangsu Jingmei Biotechnology Co., LTD, Yancheng, China, number: JM-02446M1). Tumor necrosis factor-α (TNF-α) ELISA Kit (Jiangsu Jingmei Biotechnology Co., LTD, number: JM-02415M1). Secretory immunoglobulin A (SIgA) ELISA Kit (Jiangsu Jingmei Biotechnology Co., LTD, number: JM-02723M1).

### 2.3. Animals

Male Kunming mice (Slack Jingda Experimental Animal Co, Ltd., Changsha, China, number: SCXK [Xiang] 2016-0002), aged 4 weeks and bred at the experimental animal center of Hunan University of Chinese Medicine (Changsha, China), were used for the study. The mice were housed under specific pathogen-free conditions and reared in line with the standardized methods at a temperature of 23–25 °C, a humidity of 50–70%, and a 12 h dark-light cycle with free access to food and water. The animal experiments were approved by the Animal Ethics and Welfare Committee of Hunan University of Chinese Medicine (permission number: LLBH-202106120002). To exclude the effect of sex on the gut microbiota of mice [25], only male mice were used in this study.

### 2.4. Animal Treatment

Following an accommodation period of three days, ten male mice were randomly divided into the control (CC) group and the model (CM) group, with five mice in each group. Following an improvement of the modeling method, in reference to the literature [20,22,23,26], the CM group was gavaged with the adenine suspension, 50 mg/(kg·d), 0.4 mL/each, once a day, for 14 days. From the 8th day of modeling, the CM group was gavaged with the *Folium sennae* decoction, 10 g/(kg·d), 0.4 mL/each, once a day, for 7 days. The mice in the CC group were intragastric with an equal volume of sterile water, once a day, for 14 days. Moreover, in our previous experiments, we have successfully established and verified the reliability of a diarrhea mouse model using the same modeling method [22]. The specific process was shown in Figure 1.

### 2.5. Sample Collection

Following the experiment, all mice were sacrificed by sampling orbital blood under sterile conditions. Blood samples were collected for the blood biochemistry and ELISA. Under aseptic conditions, the kidney tissues, colonic tissues and small intestine tissues of the mice were removed. The connective tissue was removed from the surface of the kidney and placed in a 4% paraformaldehyde solution for fixation for the subsequent hematoxylin and eosin (H&E) staining. The contents of the colonic tissue were washed clean with sterile water. Then, the cleaned colon tissue was placed into sterile EP tubes. Each mouse colon tissue sample was placed in a separate sterile EP tube, labelled and stored at −80 °C in the refrigerator for the subsequent ELISA. Following the removal of the mouse small intestine tissue, we collected the content samples in the small intestine tissue. The samples of the contents from each mouse were individually placed in sterile EP tubes, labeled and stored in a refrigerator at −80 °C for the subsequent 16S rRNA high-throughput sequencing [27].

### 2.6. General Behavioral Observations

We observed the behavioural status of the CC mice and the CM mice on day 14 of the modeling period. The behavioural status of mice in the CC group was used as a reference to observe the changes in the behavioural status of mice in the CM group, so as to illustrate the effect of the model of the adenine combined with *Folium sennae*-induced diarrhea on the behaviour of the mice. The mice were observed in terms of their mental state, voluntary activity, hair shape and color, faecal characteristics and anal cleanliness [28]. In addition, the body weight of the mice was measured and recorded from the 1st, 5th, 9th and 13th day of modeling.

### 2.7. Histological Observation of the Kidneys

We took out the kidney tissues placed in a 4% paraformaldehyde solution for fixation. Following the completion of the steps of the gradient ethanol dehydration, xylene transparency, paraffin embedding, sectioning and staining, the kidney histopathological changes were observed under a light microscope. The renal pathology scoring criteria [29]: 1. Glomerular pathology score: (1) Proliferation of the thylakoid cells: 0, one, two and three points according to no, mild, moderate and severe proliferations, respectively; (2) Stromal widening: normal, 0 points; mild change, no significant change in the capillaries, one point; moderate change, five capillaries (<50%) stenosis atresia, two points; severe change, capillaries (>50%) stenosis atresia, three points; (3) Degree of sclerosis: normal, 0 points; number of the sclerotic glomeruli < 25%, one point; number of the sclerotic glomeruli between 25–50%, two points; number of the sclerotic glomeruli > 50%, three points; (4) Crescent or fiber formation: none, 0 point; diffuse distribution (<25%), one point; diffuse distribution (25–50%), two points; diffuse distribution (>50%), three points. (5) Tubular and interstitial renal pathology score: (1) The area of the renal tubular atrophy was scored as 0, <25%, 25–50%, and >50%, respectively, 0, one, two and three points; (2) Lymphocytic infiltration was scored as 0, one, two and three points for no, scattered, focal and diffuse distributions, respectively; (3) The area of the interstitial fibrosis was scored as 0, <25%, 25–50%, and >50%, respectively, 0, one, two and three points.

### 2.8. Blood Biochemical Measurement

The levels of the blood urea nitrogen (BUN) and creatinine (Cr) were determined by an automatic biochemical analyser. Firstly, the machine was turned on and preheated to complete the calibration, the quality control and setting the sample serial numbers. Next, the calibrator, quality control material and test samples were loaded according to the set serial numbers. Finally, the instrument was operated to complete the calibration, the quality control measurement and the sample testing.

### 2.9. ELISA Analysis

The blood sample was left to stand for 30 min at room temperature. Following the centrifugation at 3000 r/min for 10 min, the serum was separated and loaded into sterile centrifuge tubes. According to the instructions, the sample adding, the enzyme adding, the incubation, the plate washing, the coloring, the reaction termination and the machine detection, were carried out.

The collected colon tissue samples were ground, and the supernatant was obtained. According to the instructions, the sample adding, the enzyme adding, the incubation, the plate washing, the coloring, the reaction termination and the machine detection were carried out.

### 2.10. Preparation of the Total DNA and the 16S rRNA High-Throughput Sequencing

The total genomic DNA samples were extracted from the samples of the gut contents, using the bacterial DNA Kit (OMEGA, Shanghai, China). The quantity and quality of extracted DNA were determined by a NanoDrop NC2000 spectrophotometer (Thermo Fisher Scientific, Waltham, MA, USA) and agarose gel electrophoresis. The forward primer 27F (5′-AGAGTTTGATCMTGGCTCAG-3′) and the reverse primer 1492R (5′-GGACTACHVGGGTWTCTAAT-3′) were used for the PCR amplification of the bacterial 16S rRNA gene. The 16S rRNA gene was amplified using a polymerase chain reaction (PCR) using a Q5 high-fidelity DNA polymerase (New England BioLabs, Beijing, China). The PCR products were detected using a 2% agarose gel electrophoresis and purified using a Axygen^®^AxyPrep DNA gel extraction kit. The recovered PCR amplification products were quantified through fluorescence using the Quant-it PicoGreen dsDNA Assay Kit. According to the fluorescence quantitative results, the samples were mixed in proportion to the sequencing requirements of each sample. The sequencing was completed by Paiseno Biological Co., LTD (Shanghai, China).

### 2.11. Gas Chromatography-Mass Spectrometry (GC-MS)

We detected the samples of the gut contents of fivemice in the CC group and five mice in the CM group, for acetic acid, propionic acid, butyric acid, isobutyric acid, valeric acid and isovaleric acid. The specific steps were shown below: Firstly, we took 0.1 g of acetic acid, propionic acid, butyric acid, isobutyric acid, valeric acid and isovaleric acid with a 100 mL volumetric flask and added ether to fix the volume as a reserve solution. We respectively took 1 mL, 0.75 mL, 0.5 mL and 0.25 mL of the reserve solution in a 100 mL volumetric flask, and fixed the volume with ether. Following the configuration, the sample was analyzed and the standard curve was plotted. Then, we added 2 mL of water (1:3 aqueous phosphate solution) to the samples and homogenized for 2 min. One mL of ether was added to the extract for 10 min at 4000 r/min and centrifuged for 20 min. Next, 1mL of ether was added and extracted for 10 min at 4000 r/min and centrifuged. Then, the two extracts were combined and volatilized to within 1 mL. Finally, the acetic acid, propionic acid, butyric acid, isobutyric acid, valeric acid and isovaleric acid in the gut contents of the mice were detected by the machine. The GC-MS conditions are shown in Table 1. The above testing process was completed by Qingdao Yixin Co., LTD (Qingdao, China), and the above preparation procedures were performed by the GC-MS external standard method.

### 2.12. Bioinformatics Analysis

QIIME2 and R package (V3.2.0) were used for sequence data analysis. The sequencing depth, sampling adequacy and sample homogeneity were tested by the dilution curve, the species accumulation curve and the coverage variation index to evaluate the quality of the sequence data. The ASV table of QIIME2 was used to calculate the alpha diversity index of the ASV level, such as the Chao1, Observed species, Shannon and Simpsons indexes. The beta diversity analysis used the Bray–Curtis metric to investigate the structural variation in the microbial communities between the samples and they were visualized using the principal coordinate analysis (PCoA) [30]. A random forest analysis was used to detect the groups with significant differences in the abundance of the gut content microbiota and to identify the potential biomarkers [31]. The receiver operating characteristic curve (ROC) analysis was plotted and the area under the curve (AUC) [32]. The AUC is the area under the ROC enclosed by the coordinate axis. It is a common model evaluation metric in classifiers in machine learning and can be used in the evaluation of models for various diseases as a way to predict the value of the disease models. The value of the AUC usually ranges from 0 to 1. The closer the AUC is to 1, the higher the predictive value of the disease model. The spearman correlation coefficient was calculated to construct a correlation network to explore the synergistic/competitive relationship between the different microbiota [33]. The Phylogenetic Investigation of Communities by Reconstruction of Unobserved States (PICRUSt2) is capable of predicting the functional abundance of the samples by the sequence abundance of the marker genes in the sample and is suitable for the functional prediction analysis of the gut microbiota [34]. In this study, the KEGG homologous genes and the EC enzyme classification number from the PICRUSt2 software were used for the functional prediction of the gut contentmicrobiota. The redundancy analysis (RDA) was used to investigate the association of acetic acid, propionic acid, butyric acid, valeric acid, isobutyric acid and isovaleric acid with the kidney function and the inflammation response.

### 2.13. Statistical Analysis

SPSS 24.00 software was used for the statistical analysis, and the data obtained in each group was expressed as the mean ± standard deviation. If the data of the two groups were in line with the normal distribution and homogeneity of variance, an independent sample *t* test was used. *p* < 0.05 was considered significant. *p* < 0.01 was considered extremely significant.

## 3. Results

### 3.1. Behavioral Changes in the Mice

The mice in the CC group had a normal mental state and autonomous activity, they were responsive, had smooth fur, soft and formed faeces, moderately dry and wet, and a clean perianal area. The mice in the CM group were in poor mental condition, unresponsive, had a curved and arched back, sparse and dull fur, damp bedding, a dirty perianal area and loose feces that stuck to the bedding (Figure 2). It could be seen that the adenine combined with *Folium sennae*-induced diarrhea altered the behavioral performance of the mice.

We could see that the difference in body weight between the mice in the CC group and the CM group at the 1st day of modeling was not significant. At the 5th day of modeling, the mice in the CM group showed a slight increase in body weight, compared with the CC group during the same period, but the difference was not significant. The body weight of the mice in the CM group was significantly lower (*p* < 0.01) compared with that of the CC group at the 9th day of modeling. At the 13th day of modeling, the body weight of the mice in the CM group was significantly lower compared with that of the CC group (*p* < 0.01) (Table 2). These results indicated that the adenine combined with *Folium sennae*-induced diarrhea had the suppressive effects on the body weight of mice.

### 3.2. Changes in the Kidney Structure and Function in the Mice

In the Figure 3A, the structure and morphology of the nephrons of the mice in the CC group were normal without any abnormal pathological manifestations. In the CM group, the glomerular thylakoid was hyperplastic, the interstitium was edematous and the inflammatory cells were aggregated. The renal tubules were dilated to varying degrees and the lumen was enlarged. The detailed pathological score contents were shown in Table 3. Furthermore, compared with the CC group, the serum Cr in the CM group had the increased tendency (*p* > 0.05), whereas BUN had the decreased tendency (*p* > 0.05) (Figure 3B,C), suggesting that the adenine combined with *Folium sennae*-induced diarrhea exhibited changes in the kidney structure and function in mice.

### 3.3. Analysis of the Intestinal Inflammatory Reaction in the Mice

Compared with the CC group, SIgA in the colon tissue of the mice trended downwards after the modeling (*p* > 0.05) (Figure 3D), serum IL-6 and TNF-α had the increased tendency in the CM groups (*p* > 0.05; *p* > 0.05) (Figure 3E,F). These results indicated that the adenine combined with *Folium sennae*-induced diarrhea affected the intestinal inflammatory reaction in the mice.

### 3.4. Analysis of the Structure and Function of the Gut Content Microbiota in the Mice

#### 3.4.1. Sequencing Data Quality Assessment

The experimental sequencing depth was analysed by plotting the dilution curves to check whether the sequencing volume covered all of the microbial taxa (Appendix A Appendix A). The curve flattens out as the sequencing depth increases, indicating that the sequencing depths of the two sets of samples were adequate and reasonable, covering most of the biological species and satisfying the subsequent studies.

In order to evaluate whether the samples could represent the overall situation, the species accumulation curve was drawn (Appendix A Appendix A). The curve flattened out as the number of samples increased. It demonstrated that with the addition of the new samples, the total number of OTUs almost did not increase, which proved that the sample of this study was sufficient to meet the needs of the study.

Good’s coverage index was introduced to evaluate the homogeneity of the samples within the group (Appendix A Appendix A). It could be found that Good’s coverage index of the samples in the same group was basically above 99%, which indicated that no outlier samples appeared, which proved that the samples met the needs of the experimental design. According to the results of the sequencing depth, the sampling adequacy and the sample homogeneity tests, the experimental data met the needs of experimental design and the downstream analysis.

#### 3.4.2. Analysis of the Diversity, Richness and Microbiota Structure of the Gut Content Microbiota in Mice

As shown in the Figure 4A–D, the Chao1 andthe Observed species indexes were slightly lower in the CM group than in the CC group (*p* > 0.05; *p* > 0.05), while the Simpson and Shannon indexes showed increased trends in the CM group (*p* > 0.05; *p* > 0.05), suggesting that the alpha diversity reshaped after the modeling. This study evaluated the beta diversity based on the Bray–Curtis distance algorithm (Figure 4E). The contribution rate of the abscissa PCo1 was 72.9%, and the contribution rate of the ordinate PCo2 was 9.5%. The CM samples were efficiently separated from the CC samples, and presented the phenomenon of grouping and aggregation. Apparently, the community structure of the gut content microbiota between the CC group and CM group had a distinct alteration.

#### 3.4.3. Analysis of the Dominant Bacteria of Gut Contents in the Mice

The sequences obtained above were combined and divided into OTUs, according to a sequence similarity of 97%, and the number of OTUs shared by each group was calculated. The CC group had a total of 138 OTUs with 108 unique OTUs, the CM group had a total of 159 OTUs with 129 unique OTUs, and the two groups had a total of 30 OTUs (Figure 5A). Diarrhea dramatically altered the gut content microbiota at six taxonomic levels (Figure 5B). In the taxonomic hierarchy tree diagram (Figure 5C), we found that the CC group had a larger proportion of bacteria under the level of bacteroidetes than the CM group. The proportion of bacteria under the level of proteobacteria in the CM group was more than that in the CC group. We screened the top 10 phyla, genera and species in terms of rthe elative abundance and presented them in bar charts. In the CC group and CM group, firmicutes, proteobacteria and bacteroidetes were more abundant at the phylum level (Figure 5D). At the genus level, *Candidatus Arthromitus*, *Lactobacillus*, *Burkholderia* and *Muribaculum* were more abundant (Figure 5G). The species level showed an enrichment in *Lactobacillus johnsonii*, *Lactobacillus reuteri*, *Lactobacillus intestinalis* and *Lactobacillus murinus* (Figure 5J). We used chord diagrams to summarize the dominant phyla, genus and species with an abundance greater than 1% (Figure 5E,H,K). Then, the statistical analysis of the above species revealed (Figure 5F,I,L) that, compared to the CC group, the CM group had a notably higher abundance of proteobacteria (3.00 vs. 6.54%, *p* = 0.029) and *Burkholderia* (2.49 vs. 5.92%, *p* = 0.004), *Muribaculum* (5.22 vs. 1.21%, *p* = 0.05) and *Lactobacillus reuteri* (8.07 vs. 1.63%, *p* = 2.99 × 10^−3^), manifesting a change in the composition of the dominant bacteria at the phylum, genus and species levels in the gut content microbiota of the mice after the modeling, and the major changes were proteobacteria, *Burkholderia*, *Muribaculum* and *Lactobacillus reuteri*.

#### 3.4.4. Analysis of the Characteristic Bacteria of the Gut Contents in the Mice

A nonlinear relationship between the CC and CM groups was identified using the random forest analysis to screen for the key difference groups. We constructed a random forest diagnostic model to distinguish the CM group from the CC group by using eight characteristic bacteria at the species levels (Figure 6A). We elaborated on the characteristic bacteria *Lactobacillus intestinalis* and *Bacteroides acidifaciens* that contributed most at the species level in the random forest analysis (Figure 6B). The ROC results displayed (Figure 6C,D) that *Lactobacillus intestinalis* (AUC = 0.84) and *Bacteroides acidifaciens* (AUC = 0.8) presented large AUC values, denoting that *the* modeling caused the enrichments of the characteristic bacteria *Lactobacillus intestinalis* and *Bacteroides acidifaciens*.

#### 3.4.5. Functional Analysis of the Gut Content Microbiota in the Mice

In the Figure 7A, the gut content microbiota function was generally divided into six categories, and the second level included 35 sub-functional categories, with the metabolic function accounting for a greater abundance. Among them, the gut content microbiota of the diarrhea mice had a significant role in regulating the biodegradation and metabolism of the exogenous organisms, the amino acid metabolism, carbohydrate metabolism and the energy metabolism (Figure 7B). The application of the Picrust2 software to predict the 135 KEGG homologous genes and 333 EC enzyme labels of the microbiota (Figure 7C,F). The samples in the two databases showed a significant separation (Figure 7D,G). Furthermore, the five metabolic pathways associated with the CC group and the 13 associated with the CM group in the KEGG database, were significantly different (Figure 7E). The significant differences between the 26 metabolic pathways were associated with the CC group in the MetaCyc database and the two associated with the CM group (Figure 7H). Subsequently, Cytoscape 3.7.2 was used to construct the interaction network between *Lactobacillus*
*intestinalis*, *Bacteroides*
*acidifaciens* and the metabolic functions to reflect the interaction between them, in the process of diarrhea (Figure 7I,J). *Lactobacillus intestinalis* and *Bacteroides acidifaciens* presented a negative regulation with ko01055, ko00730, ko00600, ko00521, ko00300, and a positive regulation with POLYAMSYN-PWY and HISDEG-PWY. Taken together, the metabolic functions and the pathways described above might be the main pathways affecting the changes of the gut content microbiota in mice after the modeling.

### 3.5. Changes in the Acetic Acid, Propionic Acid, Butyric Acid, Valeric Acid, Isobutyric Acid and Isovaleric Acid of Gut Contents in the Mice

Compared with the CC group, the acetic acid, propionic acid, butyric acid, valeric acid, isobutyric acid and isovaleric acid levels were notably lower in the CM group of mice (*p* < 0.01; *p* < 0.01; *p* < 0.01; *p* < 0.01; *p* < 0.01; *p* < 0.01; *p* < 0.01), suggesting that after the modeling, the production of SCFAs were enormously inhibited in the gut contents of the mice (Figure 8).

### 3.6. Correlation Analysis between the Characteristic Bacteria, SCFAs, Intestinal Inflammatory Factors and Kidney Function

We performed an intragroup correlation analysis and plotted the correlation coefficients for eight characteristic bacteria at the species levels (Figure 9A,B). Combined with the intra-group correlation coefficient analysis, we constructed the “characteristic bacteria-characteristic bacteria” interaction networks for the CC and CM groups, respectively. *Lactobacillus johnsonii* and *Lactobacillus murinus* significantly altered the regulatory role of *Lactobacillus intestinalis* in the network of the shared mutualistic relationships. The modeling resulted in the formation of specific interactions that were positively regulated by *Phascolarctobacterium faecium*, *Streptococcus thermophilus* and *Bacteroides acidifaciens* (Figure 9C,D). *Lactobacillus murinus*, *Mus musculus* and *Lactobacillus johnsonii* formed negative regulatory-specific interactions with *Bacteroides acidifaciens*, while it formed positive regulatory-specific interactions with *Lactobacillus murinus*, *Mus musculus* and *Lactobacillus johnsonii* (Figure 9E). Hence, we speculated that this might be due to the expression of the changes in the composition of the mice gut content microbiota in the diarrhea.

*Lactobacillus intestinalis* and *Bacteroides acidifaciens* showed a negative correlation with acetic acid, propionic acid, butyric acid, valeric acid, isobutyric acid and isovaleric acid. Among them, *Lactobacillus intestinalis* and *Bacteroides acidifaciens* presented a significant difference in the regulation of acetic acid (Figure 9F,G). The RDA analysis pointed out (Figure 9H) that acetic acid, propionic acid, butyric acid, valeric acid, isobutyric acid and isovaleric acid were positively correlated with SIgA. SIgA was negatively correlated with IL-6 and TNF-α. TNF-α and IL-6 were positively correlated with Cr and negatively correlated with BUN. These results suggested that the interaction among the above factors might be the mechanism of the gut-kidney impairment in the adenine combined with *Folium sennae*-induced diarrhea of the mice.

## 4. Discussion

### 4.1. Disturbed Gut Contents Microbiota in Mice Could Be an Important Factor in the Adenine Combined with Folium sennae-Induced Diarrhea

With the booming development of high-throughput sequencing in the field of gut microbiology, it has greatly facilitated and changed the knowledge of human research on the structure and function of microorganisms in the ecological context [35]. By means of bioinformatics, a diversity analysis, a dominant bacteria analysis, a random forest analysis and a functional prediction analysis, we explored the gut content microbiota of the diarrhea mice and normal mice. In the presequencing data quality assessment, we confirmed the reliability of the experimental data. The richness, diversity and community structure of the mice in the CM group were altered, which supported the theory that the gut microbiota diversity was altered in the disease states [36,37]. Next, we examined the differences between the top 10 dominant phyla, genera and species in the two groups that were ranked in relative abundance and met an abundance greater than 1%, it showed that proteobacteria and *Burkholderia* were significantly upregulated in the CM group, whereas *Muribaculum* and *Lactobacillus reuteri* were markedly downregulated. These results suggested that proteobacteria, *Burkholderia*, *Muribaculum* and *Lactobacillus reuteri* played positive or negative roles in the occurrence and development of diarrhea. The random forest analysis pointed out that *Lactobacillus intestinalis* and *Bacteroides acidifaciens* were enriched in the CM group, suggesting *Lactobacillus intestinalis* and *Bacteroides acidifaciens* might have played the important roles to affect the development of diarrhea. Moreover, in the correlation analysis between the characteristic bacteria and the metabolic pathways, we found that ko01055 showed a strong negative correlation with *Lactobacillus intestinalis* and *Bacteroides acidifaciens*. Studes have confirmed [38] that microorganisms grow to a certain stage and produce chemically complex secondary metabolites, such as antibiotics, toxins, hormones, pigments, etc. Usually they are secreted outside the cell and play an important role in the process of competition with other organisms. ko01055 involves the biosynthetic process of vancomycin and belongs to the metabolism of terpenoids and polyketides in the secondary classification of the KEGG metabolic functions. From these, we hypothesized that the characteristic bacteria *Lactobacillus intestinalis* and *Bacteroides acidifaciens* might regulate the metabolic pathways of the terpenoids and polyketides by interfering with the interactions between the metabolites and other organisms, thereby affecting diarrhea. As mentioned above, the adenine combined with *Folium sennae*-induced diarrhea was closely associated with the dysbiosis of gut content microbiota in mice. Further, the enrichments of the characteristic bacteria *Lactobacillus intestinalis* and *Bacteroides acidifaciens* played the critical roles in diarrhea.

### 4.2. Closely Related among the Characteristic Bacteria, SCFAs, Intestinal Inflammatory Response and Kidney Function

As a highly dynamic and individualized complex ecosystem, the gut miocrobiota produces metabolites SCFAs under the influence of multiple factors that produce a wide range of response effects with the numerous tissues and organs of the organism, participating in the regulation of health and disease [39,40]. The disturbed microbiota caused the abnormal production of the SCFAs, as well as the deregulation of the gut homeostasis, such as the over-activation of the intestinal immunity and the low inflammatory response states [41]. The above experimental results confirmed that the enrichment of characteristic bacteria *Lactobacillus intestinalis* and *Bacteroides acidifaciens* in the gut content microbiota of the mice, affected the changes of the bacterial community structure and function during diarrhea. Jacobson et al. [42] have shown that propionic acid produced by *Bacteroides acidifaciens* mediated the colonization resistance in enteric pathogens. Our experiments, likewise, presented a clear inhibition of acetic acid by *Bacteroides acidifaciens*. The *Lactobacillus intestinalis* is often reported as a member of the lactic acid-producing *Lactobacillus*, which enhances the host immune function [43]. In our experiments, it presented a markedly negative regulatory effect with acetic acid, and we speculated that might be related to the changes in the regulatory effect of the most characteristic bacteria of *Lactobacillus intestinalis* after the modeling. In a word, *Bacteroides acidifaciens* and *Lactobacillus intestinalis* had the correlations with the SCFAs.

The gut microbiota and its bioactive molecules interact with the immune cells and play an important role in the development of inflammation [44]. It has been reported [45] that the gut microbiota disorders damage the intestinal mucosal barrier, leading to the release of endotoxins, the activation of the local and systemic immune responses and the high levels of the inflammatory factor expression. As important pathophysiological mediators in the inflammatory process of the body, among which IL-6 and TNF-α are involved in the growth and differentiation and the functional regulation of a variety of tissue cells in the body, they are the main pathogenic inflammatory factors in the inflammatory response of the body [46]. SIgA is the main effector molecule of the intestinal mucosal immune barrier, and its secretion reduction can lead to bacterial overgrowth and directly affect the composition and function of the gut microbiota [47]. *Bacteroides acidifaciens* is the predominant *Bacteroidetes* species in the mouse intestinal microbiota that degrades mucin in the gut and induces intestinal microbial malnutrition and intestinal inflammation [48,49,50]. The correlation analysis in our experiments also confirmed the negative regulatory effect between *Bacteroides acidifaciens* and SIgA, and the positive regulatory effect between *Bacteroides acidifaciens* and TNF-α. *Bacteroides acidifaciens* showed a negative regulatory effect with IL-6, and we speculated that might be related to the changes in the interactions among the characteristic bacteria after the modeling or involve the changes in the bacterial metabolites. In addition, reports on *Lactobacillus intestinalis* are mostly seen for its protective effect on the host immune function [43]. This was consistent with our correlation results. Taken together, *Bacteroides acidifaciens* and *Lactobacillus intestinalis* had the correlations with the intestinal inflammatory response.

Studies have confirmed that Cr is one of the sensitive indicators of renal impairment. When the glomerular filtration rate decreases by more than one third, Cr increases significantly [51]. Changes in BUN depend mainly on the rate of the protein catabolism in the body and the excretory capacity of the kidneys. When the renal parenchyma is damaged, the glomerular filtration rate decreases and BUN increases [52]. Combined with the results of this experiment, there was a increasing trend of the Cr level and decreasing BUN level in the serum of mice in the CM group, compared with the CC group. This might be related to the degree of absorption of the adenine in the mice. The target organ of adenine is the kidneys, and the effect on serum is the cumulative damage after kidney failure. Additionally, BUN is the end product of the protein metabolism in the body, and the decrease in BUN may be caused by the disturbance of protein metabolism in the body after the modeling, resulting in a decrease in the rate of protein catabolism. It was found [53] that *Bacteroides acidifaciens* can be metabolized to produce indolyl sulfate, and there is a strong correlation between indolyl sulfate and kidney function. The association between *Lactobacillus intestinalis* and kidney function has been less reported in previous studies. In summary, *Bacteroides acidifaciens and Lactobacillus intestinalis* had correlations with the SCFAs, intestinal inflammation and kidney function in the adenine combined with *Folium sennae*-induced diarrhea (Figure 10). Combined with the results of the KEGG functional analysis of the gut content microbiota, we tentatively speculated that their causal association might involve the metabolic pathways of the terpenoids and polyketides. The characteristic bacteria *Bacteroides acidifaciens* and *Lactobacillus intestinalis* might cause inhibitory or catalytic reactions between the metabolites and other substances by interfering with the metabolic pathways of the terpenoids and polyketides, thus affecting the gut-kidney impairment in the adenine combined with *Folium sennae*-induced diarrhea. However, certain key substances or key links involved in this process still need to be explored further.

## 5. Conclusions

The adenine combined with *Folium sennae*-induced diarrhea altered the structure and function of the gut content microbiota in the mice, forming the enrichment of the characteristic bacteria *Lactobacillus intestinalis* and *Bacteroides acidifaciens*. Moreover, *Lactobacillus intestinalis* and *Bacteroides acidifaciens* had the correlations with acetic acid, intestinal inflammatory and kidney function. The mechanism of the gut-kidney impairment in the adenine combined with *Folium sennae*-induced diarrhea may be realized through the interactions between the characteristic bacteria and acetic acid, the intestinal inflammatory response, and kidney function. However, the causal association between them might be related to the mutual reaction between the substances under the metabolic pathway of the terpenoids and polyketides, but it still needs our further exploration.

## Figures and Tables

**Figure 1 cells-11-03261-f001:**
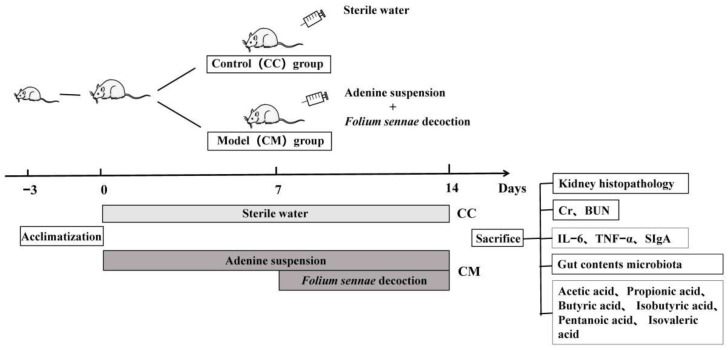
Experimental flow chart.

**Figure 2 cells-11-03261-f002:**
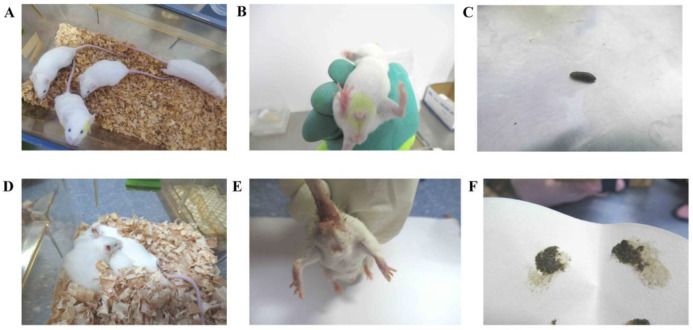
Behavioral changes in the mice. (**A**) Mental status and activity of the mice in the CC group. (**B**) Perianal condition of the mice in the CC group. (**C**) Feces of the mice in the CC group. (**D**) Mental status and activity of the mice in the CM group. (**E**) Perianal condition of the mice in the CM group. (**F**) Feces of the mice in the CM group.

**Figure 3 cells-11-03261-f003:**
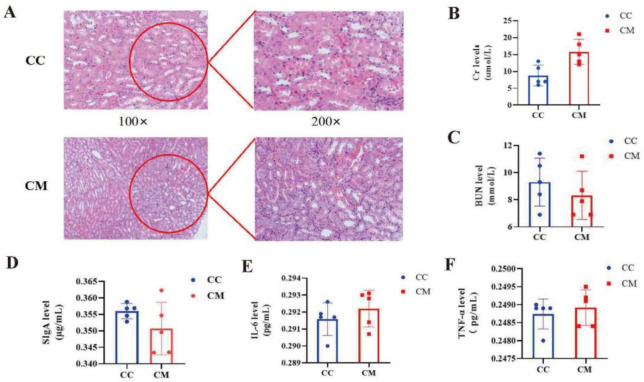
Indicators related to the kidney function and inflammatory response. (**A**) HE staining of the kidney. (**B**) Cr level. (**C**) BUN level. (**D**) SIgA level. (**E**) IL-6 level. (**F**) TNF-α level. The values were expressed as mean ± standard deviation. CC, control group (*n* = 5); CM, model group (*n* = 5).

**Figure 4 cells-11-03261-f004:**
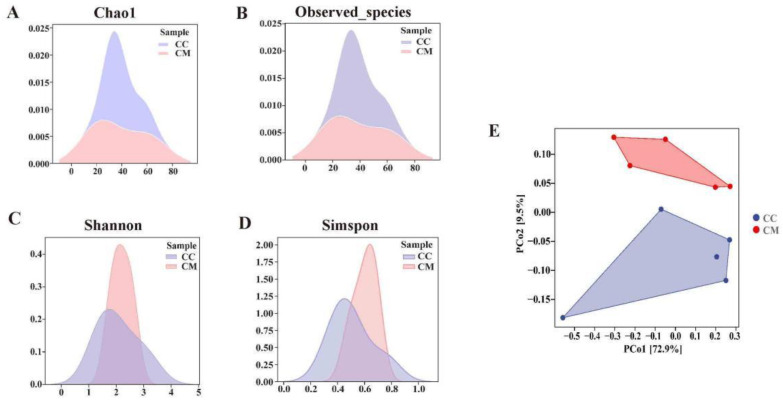
Analysis of the diversity, richness and microbiota structure of the gut content microbiota. (**A**) Chao1 index. (**B**) Observed species index. Larger Chao1 index and the Observed species index indicate higher community richness. (**C**) Simpson index. (**D**) Shannon index. Larger Simpson index and the Shannon index indicate a higher community diversity. (**E**) PCoA analysis. The closer the distance between the two groups on the axes, the more similar the community composition of the two groups in the corresponding dimension. CC, control group (*n* = 5); CM, model group (*n* = 5).

**Figure 5 cells-11-03261-f005:**
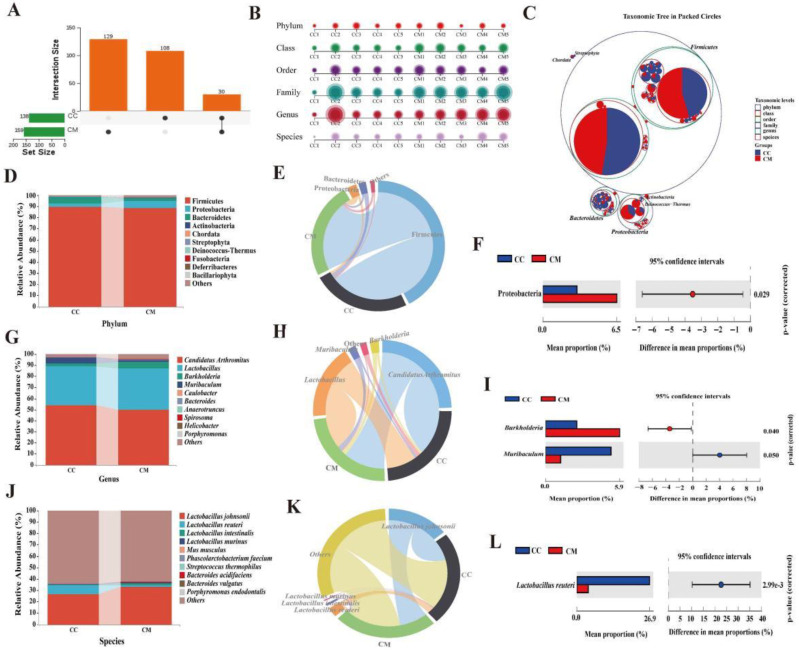
Analysis of the dominant bacteria of the gut contents in the mice. (**A**) Upset diagram. It presents the number of shared OTUs as well as the number of unique OTUs for the two groups. (**B**) Multiaxial bubble diagram. The composition of each sample in the two groups varied at six taxonomic levels. (**C**) Classification hierarchy tree diagram. Different colored circles represent different taxonomic levels. The larger the area of the sector in the circle, the higher the abundance of that taxonomic unit in the corresponding grouping. (**D**) Horizontal bar diagram of the phylum level. The top 10 dominant phyla in terms of the relative abundance. (**E**) Chord chart of the phylum level. The dominant phyla in the top 10 relative abundance rankings and greater than 1%. (**F**) Dominant phyla with a significant variation. (**G**) Horizontal bar diagram of the genus level. The top 10 dominant genera in terms of the relative abundance. (**H**) Chord chart of the genus level. The dominant genera in the top 10 relative abundance rankings and greater than 1%. (**I**) Dominant genus with significant variation. (**J**) Horizontal bar diagram of the species level. The top 10 dominant species in terms of the relative abundance. (**K**) Chord chart of the species level. The dominant species in the top 10 relative abundance rankings and greater than 1%. (**L**) Dominant species with a significant variation. CC, control group (*n* = 5); CM, model group (*n* = 5).

**Figure 6 cells-11-03261-f006:**
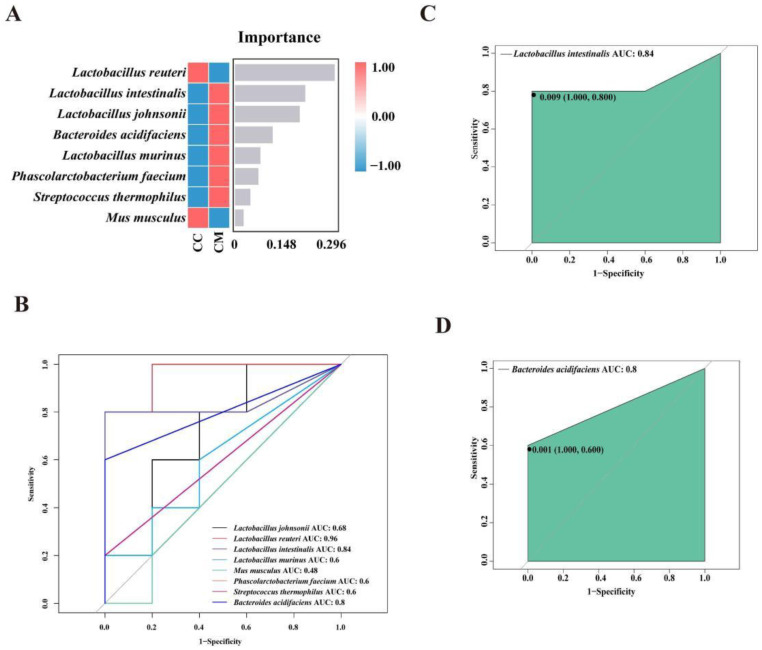
Analysis of the characteristic bacteria of the gut contents in the mice. (**A**) Random forest diagram of the species level. The abscissa is the importance scores of the species for the model and the ordinate are the names of the taxonomic units at the species level. The heat map shows the distribution of the abundance of these species in each grouping. (**B**) ROC of the species level. (**C**) ROC of *Lactobacillus intestinalis*. (**D**) ROC of *Bacteroides acidifaciens*. The ROC can be used to evaluate the biomarkers. The larger the area under the curve, the higher the diagnostic accuracy. CC, control group (*n* = 5); CM, model group (*n* = 5).

**Figure 7 cells-11-03261-f007:**
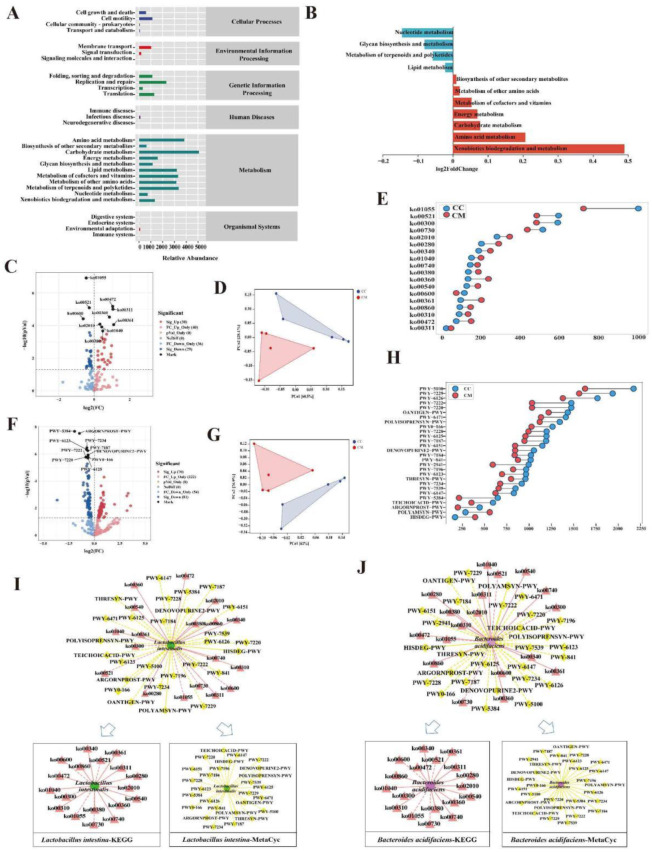
Functional analysis of the gut content microbiota in mice. (**A**) Predicted abundance of the KEGG function. The abscissa is the abundance of the metabolic functions, the ordinate is the metabolic function at the second level of the classification, and the right-hand side is the first level of classification to which the function belongs. (**B**) Histogram of the metabolic function in the positive and negative coordinates. (**C**) KEGG pathway volcanoes. Red circles represent the upregulated KEGG pathways, blue circles represent the downregulated KEGG pathways. (**D**) PCoA diagram of the KEGG functional units. (**E**) Dumbbell diagram of the KEGG pathway with significant differences. (**F**) MetaCyc pathway volcanoes. Red circles represent the upregulated MetaCyc pathways, blue circles represent the downregulated MetaCyc pathways. (**G**) PCoA diagram of the MetaCyc functional units. (**H**) Dumbbell diagram of the MetaCyc pathway with significant differences. (**I**) Interaction network of the “Characteristic bacterium *Lactobacillus intestinalis*-metabolic pathway”. (**J**) Interaction network of the “Characteristic bacterium *Bacteroides acidifaciens*-metabolic pathway”. The solid line represents the positive correlation and the dashed line represents the negative correlation. CC, control group (*n* = 5); CM, model group (*n* = 5).

**Figure 8 cells-11-03261-f008:**
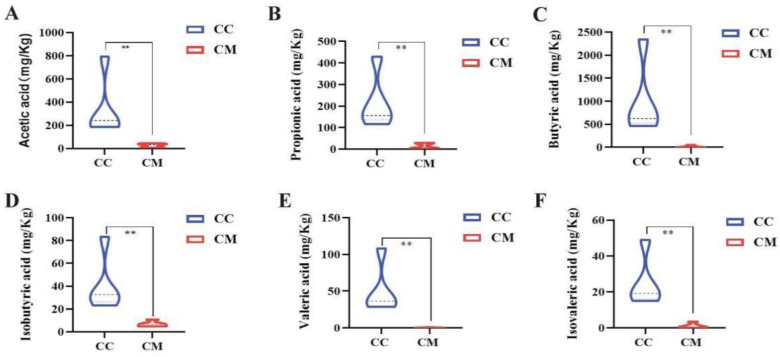
Analysis of SCFAs. (**A**) acetic acid. (**B**) propionic acid. (**C**) butyric acid. (**D**) valeric acid. (**E**) isobutyric acid. (**F**) isovaleric acid. CC, control group (*n* = 5); CM, model group (*n* = 5). The values were expressed as mean ± standard deviation. ** *p* < 0.01.

**Figure 9 cells-11-03261-f009:**
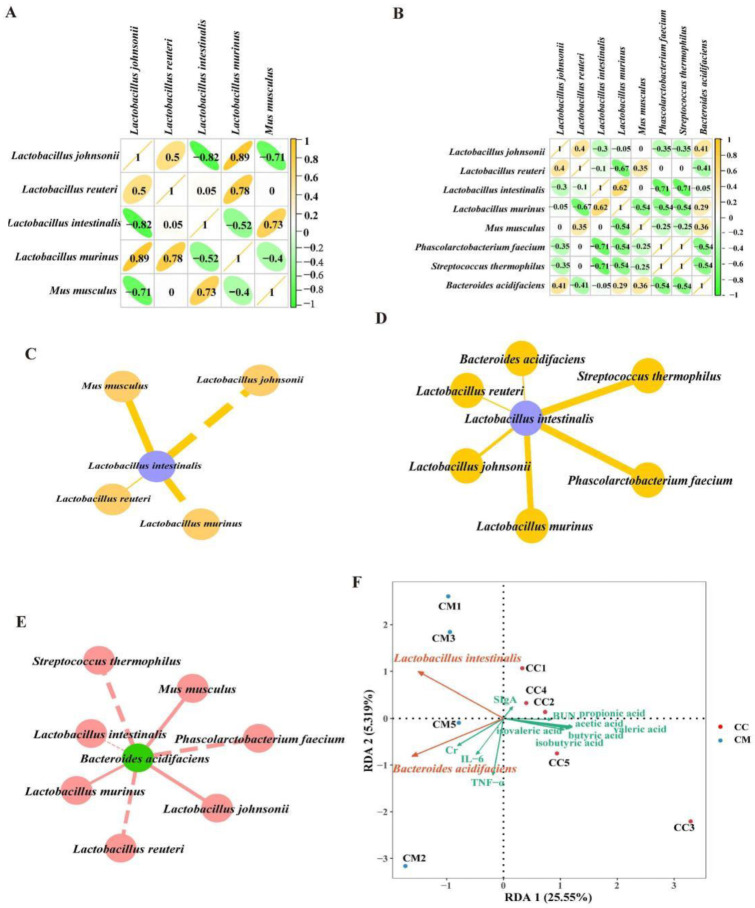
Correlation coefficient analysis of the characteristic bacteria. (**A**) Correlation coefficient diagram of the characteristic bacteria at the species level in the CC group. (**B**) Correlation coefficient diagram of the characteristic bacteria at the species level in the CM group. It presents the correlation coefficients between the characteristic bacteria within each group at the species level, with green circles representing the negative correlations and the yellow circles representing the positive correlations. (**C**) “*Lactobacillus intestinalis*-characteristic bacteria” interaction network in the CC group. (**D**) “*Lactobacillus intestinalis*-characteristic bacteria” interaction network in the CM group. (**E**) “*Bacteroides acidifaciens*-characteristic bacteria” interaction network in the CM group. The solid line represents the positive correlation and the dashed line represents the negative correlation. (**F**) Association diagram of *Lactobacillus*
*intestinalis* and the SCFAs. (**G**) Association diagram of *Bacteroides acidifaciens* and the SCFAs. (**H**) RDA. The angle between the connecting arrows represents the correlation, with an acute angle indicating a positive correlation and an obtuse angle indicating a negative correlation. CC, control group (*n* = 5); CM, model group (*n* = 5).

**Figure 10 cells-11-03261-f010:**
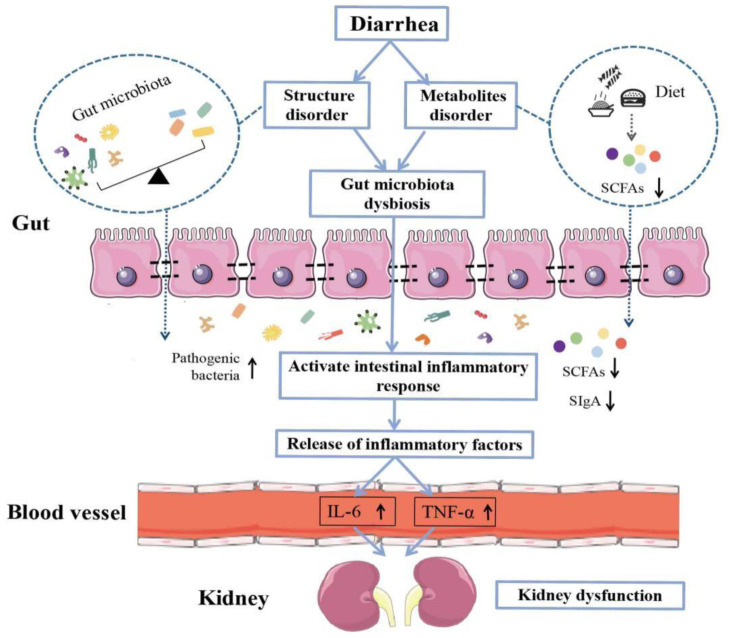
Interaction between the “bacteria-SCFAs-inflammation-kidney function” during the progress of the gut-kidney impairment in the adenine combined with *Folium sennae*-induced diarrhea.

**Table 1 cells-11-03261-t001:** GC-MS conditions.

Steps	Conditions
Column temperature requirement	100 °C (5 min)-5 °C/min-150 °C (0 min)-30 °C/min-240 °C (30 min)
Flow rate requirements	1 mL/min
Shunt ratio	75:1
Carrier gas	Helium
Chromatographic column	TG WAX 30 m × 0.25 mm × 0.25 μm
Injector	240 °C
Mass spectrometry EI source, bombardment voltage	70 eV
Single ion scan mode	Quantitative ion 63, 70
Ion source temperature	200 °C
Connection line temperature	250 °C

**Table 2 cells-11-03261-t002:** Body weight (g) of the mice.

Group	The 1st Day of Modeling	The 5th Day of Modeling	The 9th Day of Modeling	The 13th Day of Modeling
CC	24.73 ± 0.49	31.31 ± 1.34	35.25 ± 2.21	38.95 ± 2.18
CM	26.10 ± 0.81	31.61 ± 0.92	34.10 ± 1.00 **	34.03 ± 0.55 **

Note: The values were expressed as mean ± standard deviation (*n* = 5). Compared with the CC group, ** *p* < 0.01. CC, control group; CM, model group.

**Table 3 cells-11-03261-t003:** Renal pathological score of the mice.

Group	Glomerular Pathology Score	Tubular and Interstitial Renal Pathology Score
CC	0.67 ± 0.47	0.67 ± 0.47
CM	4.67 ± 0.47 *	3.67 ± 0.47 *

Note: Compared with the CC group, * *p* < 0.05. CC, control group; CM, model group.

## Data Availability

The data underlying this study is available within the manuscript. The gut content microbiota sequencing data has been uploaded to the NCBI database (https://www.ncbi.nlm.nih.gov/, accessed on 31 March 2023), no. PRJN812747.

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
