# Peer review of "Gut-Kidney Impairment Process of Adenine Combined with Folium sennae-Induced Diarrhea: Association with Interactions between Lactobacillus intestinalis, Bacteroides acidifaciens and Acetic Acid, Inflammation, and Kidney Function"

_cells, 2022, doi:10.3390/cells11203261_

Round 1
Reviewer 1 Report
Xiaoya Li et al have presented very interesting work and focused on evidence that suggests interaction of the gut microbiota with renal impairment and pathogenesis of the disease. There are very few works that have been done in this area and authors are exceptional ones. They have figured all the possible models to prove their hypothesis. However, different pathways involved in gut-kidney impairment needs to be explored further. In addition, this paper is easy to read , understand and drawn in a best presentation of the authors work and this paper could be accepted in its present form. My best wishes for them for future studies.Author Response
Dear reviewer,
Thank you very much for your review of the manuscript. We have made some modifications to the manuscript and have marked in red for your easy viewing. Many thnnks.
Best wishes,
Zhoujin Tan
Reviewer 2 Report
Li X. et al aimed to explore the characteristics of the microbiota of diarrhea and to analyze the correlation among bacteria, SCFAs, inflammatory response, and kidney function. I think their purpose is worth being pursued. However, there are significant issues in this study.
1) The authors mentioned the change of microbiota and the SCFAs is a crucial factor in the development of diarrhea. However, there are several types of diarrhea such as bacterial infection, viral infection, food intolerance, and food allergy. In this study, they used the diarrhea model with folium sennae. This kind of drug-induced diarrhea would not be a clinical problem as it is improved by drug withdrawal while bacterial infection and viral infection would be significant problems. They mentioned this study will provide ideas for clinical gut-kidney-related gut microbiota therapy for diarrhea. The authors may need to use other types of diarrhea models for their purpose.
2) To elucidate the diarrhea-kidney interaction, they should add the mouse model with only adenine and the model with only folium sennae.
3) There are no significant changes in Cr and BUN in this model. It would be doubtful that the model is appropriate as the renal failure model.
4) The authors should show the body weight, the amount of food intake, and the urine volume of the mice as diarrhea often cause dehydration.
5) The meaning of AUC values in Fig. 6 is not clear. I’m wondering what kind of disease they aim to diagnose clinically.
Minor issues
1) Although they cited a few reports describing the correlation between SCFAs and renal failure, there have been a lot of papers about the relationship. They should mention them in the introduction.
2) Preliminary experiments would be in the result section, not in the introduction.
3) The authors should show the quantitative data about the behavioral performance of mice with diarrhea.
4) There is supposed to be a period in line 331.
5) Fig. 3 and Fig. 4 would be in the method section or supplemental figures.
6) As for Fig. 7, the authors just described the results. It may need a deep discussion.
7) Although they claimed that the results might show the mechanism of gut-kidney impairment in adenine combined with folium sennae-induced diarrhea of mice, the relation between cause and effect is unclear in their data. They should add more discussion or comments about it.
Author Response
Dear reviewer,
Thanks for your guiding suggestions on the manuscript. We have revised the contents based on your suggestions. Besides, we have marked the revised parts in red for your easy viewing. The specific responses are as follows:
Major issues:
1)Thanks for your instructive suggestions. In this experiment, we study the deficiency kidney-yang syndrome, one of the six syndrome types of diarrhea in Traditional Chinese Medicine (TCM) clinical practice. There aremany factors causing the diarrhea with deficiency kidney-yang syndrome. We explored the formation mechanism of this syndrome by simulating a mouse model of diarrhea with deficiency kidney-yang syndrome through compound factor modeling. Because of the complexity of the factors involved in the formation of the different syndrome types of diarrhea, we were unable to examine each factor. In this study, we only investigated the formation mechanism of one of the symptoms of diarrhea. Additionally, we have revised some of the expressions in the manuscript in order to be more relevant to the theme of the manuscript.
2)Thanks for your instructive suggestions. In the preliminary experiments, we investigated the effects of different doses of adenine modeling on the kidney function and gut microbiota of mice, analyzed the correlation between gut microbiota and kidney function during modeling, and elucidated the role of gut microbiota in the process of kidney dysfunction after adenine modeling. Among them, adenine (50 mg/(kg·d) for 14 days by gavage) damaged the kidney structure and function of mice, and caused disorders of gut microbiota, and characteristic bacterium Lactobacillus hamsteri affected the process of kidney functional impairment. Therefore, adenine-induced kidney impairment is not an independent event. There is an association between kidney impairment caused by adenine modeling and disturbance of the gut microbiota. Besides, our research group preliminary research presentedthat mice showed obvious diarrhea symptoms and caused gut microbiota disorder after folium sennae modeling [20]. Futhermore, we compared the effects of adenine combined with folium sennae at different doses and days on kidney and intestinal function in mice and presented that adenine (50 mg/(kgd), gavaged for 14 days) combined with folium sennae (10 g/(kg·d), gavaged for 7 days) significantly caused impairment of kidney and gut function in mice [21]. Subsequently, we have successfully constructed and validated a mouse model of diarrhea using the same modeling method described above, thus confirming the reliability of the model [22]. The present study is a further research investigation based on the previous experiments. We related about the contents of the preliminary study in the “introduction” section of the manuscript ( Lines 78-98).
[20] Zhang, C.Y.; Shao, H.Q.; Li, D.D.; Xiao, N.Q.; Tan, Z.J. Role of tryptophan-metabolizing microbiota in mice diarrhea caused by Folium sennae extracts. BMC Microbiol 2020, 20, 185
[21] Li, X.Y.; Zhu, J.Y.; Wu, Y.; Tan, Z. J. Correlation Between Kidney Function and Intestinal Biological Characteristics of Adenine and Folium Sennae-Induced Diarrhea Model in Mice. Turkish J Gastroenterol 2022. doi: 10.5152/tjg.2022.211010
[22] Li, X.Y.; Zhu, J.Y.; Wu, Y.; Liu, Y.W.; Hui, H.Y.; Tan, Z.J. Model Building and Validation of Diarrhea Mice with Kidney-yang Depletion Syndrome. J Tradit Chin Med 2022, 63, 1368-1373.
3)Thanks for your instructive suggestions. In this experiment, we used adenine combined with folium sennaeto construct a diarrhea model and observed the effects on the kidney structure and function of mice, not to construct the renal failure model. Combined with kidney pathology and blood biochemical indexes in mice, we found that modeling caused structural damage and altered kidney function in mice.
4)Thanks for your instructive suggestions. In our previous paperson the construction and evaluation of the diarrhea model of deficiency kidney-yang syndrome ("Li, X.Y.; Zhu, J.Y.; Wu, Y.; Liu, Y.W.; Hui, H.Y.; Tan, Z.J. Model Building and Validation of Diarrhea Mice with Kidney-yang Depletion Syndrome. J Tradit Chin Med 2022, 63, 1368-1373. doi: 10.13288/j.11-2166/r.2022.14.012"、"Zhu, J.Y.; Li, X.Y.; Deng, N.; Peng, X. X.; Tan, Z.J. Diarrhea with deficiency kidney-yang syndrome caused by adenine combined with Folium senna was associated with gut mucosal microbiota. Front Microbiol. 2022, 13, 1007609. doi: 10.3389/fmicb.2022.1007609"), we have analyzed the body weight and food intake of mice to verify the reasonableness and reliability of the model. This study was mainly to investigate the formational mechanism of the diarrhea with deficiency kidney-yang syndrome, so there were no analysis of the body weight and food intake of mice in the manuscript. In this study we did not detect the urine volume of mice. Only the changes in behavioral performance of the two groups of mice were observed, and the results showed that the bedding material of the model group was relatively wet. Therefore, we speculated that adenine combined with folium senna might cause the phenomenon of increased urine volume in the mice after modeling. However, we did not quantify this indicator, which was a shortcoming in our experiment. Thank you very much for your guiding suggestions, and in the future we will judge the indicator by quantifying them in a more convincing way.
5)Thanks for your instructive suggestions. We have added relevant contents in the methods section (Lines 268-273).
Minor issues:
1)Thanks for your instructive suggestions. We have citedthe relevant reference confirming the relationship between SCFAs and renal function in the second paragraph of the introduction (Lines 69-73).
2)Thanks for your instructive suggestions. We citedthe previous research results of the research group in order to present the conclusions we have obtained, which provided the basis for the further research of this experiment. There is a progressive correlation between these prior experiments and our subsequent experiments.
3)Thanks for your instructive suggestions. In our previous paperson the construction and evaluation of the diarrhea model of deficiency kidney-yang syndrome ("Li, X.Y.; Zhu, J.Y.; Wu, Y.; Liu, Y.W.; Hui, H.Y.; Tan, Z.J. Model Building and Validation of Diarrhea Mice with Kidney-yang Depletion Syndrome. J Tradit Chin Med 2022, 63, 1368-1373. doi: 10.13288/j.11-2166/r.2022.14.012"), we have analyzed the body weight、anal temperature、diarrhea index and other relevant indicators of mice to verify the reasonableness and reliability of the model. This study was mainly to investigate the formational mechanism of the diarrhea with deficiency kidney-yang syndrome, so there were no analysis in the manuscript.
4)Thanks for your instructive suggestions. We have added a period and marked in red (Line 332).
5)Thanks for your instructive suggestions. The results presented in Fig.3 are the assessment of the sequencing quality of the mouse gut microbiota, which provides quality assurance for the results of the subsequent analysis. We think it is appropriate to put it in the supplementary figure. However, 4 is a subsequent demonstration of the diversity of the mouse gut microbiota and we think it is appropriate to place it in the results section.
6)Thanks for your instructive suggestions. Combined with the results of Fig. 7, we have added the discussions (lines 537-548).
7)Thanks for your instructive suggestions. We have added the possible causal links involved between them in the discussion section (lines 612-619).
Thanks for your review of the manuscript.
Best wishes,
Zhoujin Tan

Round 2
Reviewer 2 Report
Most of the author’s responses are reasonable. However, even if they have previously reported the data of this mouse model, the physiological data are usually variable. The physiological data in Fig. 2 are insufficient. They would still need to show at least the body weight. The images in Fig. 2A can be quantitated. As for result 3.1, it is not scientifically appropriate that just the author’s impression was described as the result. They should show the quantitative data or at least pictures of the mice for this part.Author Response
Dear reviewer,
Thanks for your guiding suggestions on the manuscript. We have added the body weight of the mice. The results in Fig. 2A were quantitatively analyzed and supplemented with pictures of mice. Besides, we have marked the revised parts in red for your easy viewing. Mank thanks.
Best wishes,
Zhoujin Tan
